# Fuzzy Algebras of Concepts

**Manuel Ojeda-Hernández** , **Domingo López-Rodríguez** and **Pablo Cordero** *

Departamento de Matemática Aplicada, Universidad de Málaga, Andalucía Tech, 29071 Málaga, Spain
* Correspondence: pcordero@uma.es

**Abstract:** Preconcepts are basic units of knowledge that form the basis of formal concepts in formal concept analysis (FCA). This paper investigates the relations among different kinds of preconcepts, such as protoconcepts, meet and join-semiconcepts and formal concepts. The first contribution of this paper, is to present a fuzzy powerset lattice gradation, that coincides with the preconcept lattice at its 1-cut. The second and more significant contribution, is to introduce a preconcept algebra gradation that yields different algebras for protoconcepts, semiconcepts, and concepts at different cuts. This result reveals new insights into the structure and properties of the different categories of preconcepts.

**Keywords:** formal concept analysis; graded subsethood; preconcept; protoconcept; fuzzy

**MSC:** 06B75; 06A75; 03E72; 06D72; 08A72

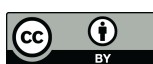

## 1. Introduction

Formal concept analysis (FCA) is a mathematical theory and tool, for analysing and discovering the relationships within data, by modeling knowledge structures and concepts. It was first introduced in the early 1980s, by Bernhard Ganter and Rudolf Wille [1], and has since become an important field of research in the areas of knowledge representation, cognitive science, data analysis, and information science. FCA is based on the mathematical theory of lattices and provides a way to formally describe and study the relationships between objects and attributes in a formal context. The importance of FCA lies in its ability to identify and categorise patterns and structures in data, making it a valuable tool for knowledge representation and management. The theory and tools of FCA have a wide range of applications, from knowledge management and information retrieval, to machine learning and artificial intelligence. FCA has been used to analyse data in a variety of domains, including, but not limited to, biology, medicine, marketing, and finance, making it an important tool for interdisciplinary research [2].

Philosophically, FCA grows in parallel with the doctrine of conceptual knowledge systems, establishing a formal model in which conceptual knowledge can be represented, acquired, inferred, and communicated [3]. In this sense, FCA incorporates the three basic notions of conceptual knowledge: objects, attributes, and concepts, which in turn are related by four basic relationships: each object has attributes, each object belongs to a concept, an attribute abstracts from a concept, and each concept is a specialisation of a more general concept [4].

The formalisation of these elements within FCA is as follows. The basic notion is that of a formal context, represented by $\mathbb{K} = (G, M, I)$, where $G$ is a set whose elements are called *objects*, $M$ is a set of *attributes*, and $I \subseteq G \times M$ denotes a relation between the elements of $G$ and $M$.

The idea of a concept in FCA, expresses a biunivocal relation between a set of objects and their precise description by a set of attributes. Thus, a (formal) concept is a pair $(A, B)$, where the set of attributes shared by the objects in $A$ is exactly the set $B$, and the only objects

that have all the attributes in $B$ are those in $A$. In general, these relations are expressed in terms of the operators $\uparrow\colon G \to M$ and $\downarrow\colon M \to G$, defined by

$$A^{\uparrow} := \{m \in M : g \: I \: m \text{ for all } g \in A\},$$

$$B^{\downarrow} := \{g \in G : g \: I \: m \text{ for all } m \in B\},$$

for sets $A \subseteq G$, $B \subseteq M$. These two operators, named *intent* and *extent*, respectively, form a Galois connection and, therefore, their compositions are closure operators. Hence, a formal concept is a pair $(A, B)$, such that $A = B^{\downarrow}$ and $B = A^{\uparrow}$.

In the development of a conceptual knowledge system, it is useful to make the notion of concept more flexible, in order to increase the expressiveness of the system, thus making it possible to represent a greater number of situations and relations than those expressible by means of the strict relation that defines a concept. Thus, in FCA, notions such as semiconcept, protoconcept, or preconcept emerge [3], which relax the conditions but maintain representativeness within the conceptual scheme. Concretely, it can be shown that generalisations such as protoconcepts and semiconcepts, can be effectively used to describe formal concepts, and that they can be operated on by means of Boolean operations, that endow them with the structure of double Boolean algebras [5]. Therefore, preconcepts, protoconcepts, and semiconcepts can be considered as different levels of *units of description* of concepts [6].

In the formal definition of these entities, we find conditions expressed as relations between $A$ and $B$. While this strategy provides generalisations of the idea of a concept, this contrasts with the fact that these relations are crisp in nature, and do not allow us to determine the degree of generalisation or, equivalently, the degree of closeness of these entities to a formal concept. Moreover, in real situations, it is difficult to find the precise correspondence between $A$ and $B$ that would allow us to ensure that $(A, B)$ is a concept, but we can wonder to what degree this pair verifies the relation and, therefore, has properties similar to that of being a concept.

In this paper, we will work with a fuzzy extension of the above ideas. Let us consider a complete residuated lattice $L$, as the structure of membership degrees, hence the incidence relation is formalised now as $I \in L^{G \times M}$, and the concept-forming operators become, for $A \in L^{G}$, $B \in L^{M}$:

$$
\begin{aligned}
A^{\uparrow}(m) &:= \bigwedge_{g \in G} (A(g) \to I(g, m)), \\
B^{\downarrow}(g) &:= \bigwedge_{m \in M} (B(m) \to I(g, m)).
\end{aligned}
\tag{1}
$$

Observe that when $L = \{0, 1\}$, these fuzzy operators coincide with their crisp versions, therefore, for the sake of simplicity, we will denote both of them by simple arrows.

The aim of this work is to propose a fuzzy framework in which the different notions of preconcept, protoconcept, semiconcept, and formal concept are unified, making them correspond to different (fuzzy) degrees of generalisation. To this end, we construct a fuzzy lattice and a fuzzy algebra over the powerset of subsets of $G$ and $M$, where the different $\alpha$-cuts indicate the *degree of closeness to the concepts*, and, in significant cases, correspond precisely to the preconcepts, protoconcepts, semiconcepts, and formal concepts. This provides a natural way of incorporating uncertainty and imprecision into the analysis, thus enabling us to model more realistic scenarios.

Recently, Šostak et al. [7] have presented an approach to the fuzzy grading of preconcepts, based on *quantales* (complete lattices equipped with an associative *product* operation, distributive with respect to arbitrary joins). The preconcept terminology in that work, does not coincide with the traditional one in FCA [8], although it is consistent with their proposal: for the authors, the set of preconcepts coincides exactly with the powerset of $G \times M$, and the *pre-* prefix only indicates that these elements are used as generalisations of formal concepts. In that work, although the designed grading is thoroughly treated, the

relationship between the different conceptual structure algebras is not explained, a problem that we address in our proposal.

From a theoretical point of view, in order to *fuzzify* the conditions that define these items, which are based on inclusion relations between sets and will be described in detail in Section 2.1, a fuzzy extension of the inclusion of sets, based on the graded subsethood of Bělohlávek [9], is used. Moreover, this choice maintains the interpretability of the classical definitions in this new fuzzy scheme.

In order to thoroughly analyse the fuzzy approach to this topic, it is important to consider various technical aspects that arise. Of particular note is the fact that while preconcepts and concepts possess a complete lattice structure, with appropriate ⊓ and ⊔ operations, protoconcepts and semiconcepts do not necessarily possess this structure. However, they do have the double Boolean algebra structure, which implies that the fuzzy extension between preconcepts and concepts must be relaxed to that of an *L*-fuzzy algebra, rather than an *L*-fuzzy lattice.

The specific main contributions of this work are:

- The creation of a gradation of the powerset lattice as an *L*-fuzzy lattice, that recovers the complete lattice of preconcepts for a specific case.
- The construction of an *L*-fuzzy algebra of preconcepts that preserves the subalgebra structure and recovers the algebra of protoconcepts for a specific case. We also extend this gradation to include both types of semiconcepts as well.
- The definition of an $L \times L \times L$-fuzzy algebra of preconcepts, that recovers the algebras of protoconcepts, semiconcepts, and formal concepts for different combinations of cuts.

The contributions of this work are therefore significant, as they provide a new and more flexible framework for analysing complex data structures, and are likely to have a significant impact on the development of new methods and techniques for data analysis, and could lead to new insights both from a theoretical and practical point of view.

This paper is structured as follows: in Section 2, we find the fundamental ideas, from conceptual structures (Section 2.1), *L*-fuzzy lattices and algebras (Section 2.2), and graded subsethood (Section 2.3). In Section 3, we present the fuzzy theoretical framework that extends the different generalisations of formal concepts, and in Section 4, the final remarks and future work that can be derived from this paper are detailed.

## 2. Preliminaries

In this section, we present the main results and ideas that serve as a starting point for our development. In what follows, as mentioned in the introduction, we will consider the symbol *L* to denote a poset which has also the structure of a complete residuated lattice [9]. If no confusion arises, we will use $\leq$ to denote its order and $\vee$ and $\wedge$ its supremum and infimum operators.

Along these preliminary notions, the concept of *fuzzy set inclusion* will be mentioned very often, so it becomes necessary to recall that, in Zadeh's sense [10], given two sets $A, B \in L^X$, we say that $A \subseteq B$ if, and only if, $A(x) \leq B(x)$ for all $x \in X$.

### 2.1. Algebraic Conceptual Structures

In the Introduction, it has been mentioned that some previous work [7] has dealt with the issue of concept grading, although without taking into account intermediate structures of preconcepts, protoconcepts, or semiconcepts. Although preconcepts are mentioned in that work, in reality this terminology refers to the powerset $L^G \times L^M$. To set the terminology, in our work, we will refer to this set as the powerset, to avoid confusion with the terms used within FCA.

The notion of preconcept used in FCA, originates from Piaget's cognitive psychology, and is related to the stage of development in which one transitions from sensorimotor intelligence to operational intelligence [11]. To formalise this idea, Wille [8], Stahl and Wille [12], and Ganter and Wille [2] define a *formal preconcept* of a context $\mathbb{K} = (G, M, I)$, as a pair

$(A, B) \in 2^G \times 2^M$ with $A \subseteq B^\downarrow$, which is equivalent to $B \subseteq A^\uparrow$. The set $\mathfrak{V}(\mathbb{K})$ of all preconcepts, equipped with the order $\leq$ given by

$$(A_1, B_1) \leq (A_2, B_2) \iff A_1 \subseteq A_2 \text{ and } B_1 \supseteq B_2,$$

is a completely distributive complete lattice, called the *preconcept lattice*. Note that this same order $\leq$ endows the powerset $2^G \times 2^M$ with a lattice structure. In both cases, in the powerset and in the preconcept lattice, the *supremum* and *infimum* operators, denoted by $\vee$ and $\wedge$ if no confusion arises, are defined by:

$$\bigwedge_t (A_t, B_t) = \left( \bigcap_t A_t, \bigcup_t B_t \right) \quad \text{and} \quad \bigvee_t (A_t, B_t) = \left( \bigcup_t A_t, \bigcap_t B_t \right), \tag{2}$$

for an arbitrary collection $\{(A_t, B_t)\}$ of elements of these sets. As stated in [13], in order to change a preconcept into a concept, one may extend each of the sets $G$ and $M$ by one element, with the appropriate incidences, which is consistent, too, with the terminology used. It can be shown, and it is known as the *Basic Theorem of Preconcept Lattices* [14], that the set of preconcept lattices is isomorphic to the set of (completely distributive) complete lattices, where the supremum of its atoms is equal or greater than the infimum of its coatoms. This definition can be naturally extended to the fuzzy setting, by letting $(A, B) \in L^G \times L^M$ satisfy $A \subseteq B^\downarrow$.

It can be verified that, if the set of preconcepts is defined with the order $\subseteq^2$ ($\subseteq$ in each of the components), the maximal elements correspond precisely with the formal concepts. In addition, a formal preconcept $(A, B)$ is capable of generating two concepts, using the Galois connection defined in the formal context, since $(A^{\uparrow\downarrow}, A^\uparrow)$ and $(B^\downarrow, B^{\downarrow\uparrow})$ are formal concepts in $\mathbb{K}$. Thus, a preconcept can be thought of as a formal concept of a subcontext of $\mathbb{K}$.

An alternative viewpoint studies this set of preconcepts with algebra structure. We can define logical operations on $\mathfrak{V}(\mathbb{K})$ as follows:

$$
\begin{aligned}
(A_1, B_1) \sqcap (A_2, B_2) &:= \left( A_1 \cap A_2, (A_1 \cap A_2)^\uparrow \right), \\
(A_1, B_1) \sqcup (A_2, B_2) &:= \left( (B_1 \cap B_2)^\downarrow, B_1 \cap B_2 \right), \\
\neg (A, B) &:= \left( G \setminus A, (G \setminus A)^\uparrow \right), \\
\lrcorner (A, B) &:= \left( (M \setminus B)^\downarrow, M \setminus B \right), \\
\bot &:= (\varnothing, M), \\
\top &:= (G, \varnothing),
\end{aligned}
\tag{3}
$$

such that $\underline{\mathfrak{V}}(\mathbb{K}) = (\mathfrak{V}(\mathbb{K}), \sqcap, \sqcup, \neg, \lrcorner, \bot, \top)$ is a *generalised double Boolean algebra* called the *preconcept algebra*. Observe that, in this terminology, the structure of double Boolean algebra is neither a generalisation nor a particularisation of the lattice structure: $L^G \times L^M$ is a complete lattice but it is not a double Boolean algebra with this operation, and not all double Boolean algebras are lattices, as seen in the following example.

**Remark 1.** *Notice that these operations $\sqcap$ and $\sqcup$ do not provide $\mathfrak{V}(\mathbb{K})$ with a lattice structure. For example, consider the formal context $\mathbb{K} = (G, M, I)$ given by $G = \{u\}, M = \{x_1, x_2\},$ $I(u, \cdot) = (0, 1),$ where we consider $L = [0, 1]$ equipped with the Gödel structure. Then, the pair $(\{u\}, \varnothing)$ is a preconcept, since $\varnothing \subseteq \{u\}^\uparrow = \{x_2\}$. However,*

$$(\{u\}, \varnothing) \sqcap (\{u\}, \varnothing) = (\{u\}, \{u\}^\uparrow) = (\{u\}, \{x_2\}).$$

*Therefore, the operation $\sqcap$ is not idempotent and cannot be the infimum of a lattice.*

*Similarly, for* $(\varnothing, \varnothing)$ *we have,*

$$(\varnothing, \varnothing) \sqcup (\varnothing, \varnothing) = (\varnothing^{\downarrow}, \varnothing) = (\{u\}, \varnothing).$$

*Therefore, the operation $\sqcup$ is not idempotent and cannot be the supremum of a lattice.*

**Proposition 1** ([5,15])**.** *In the preconcept algebra $\underline{\mathfrak{V}}(\mathbb{K})$, the following equations are valid:*

$$
\begin{aligned}
(x \sqcap x) \sqcap y &= x \sqcap y & (x \sqcup x) \sqcup y &= x \sqcup y \\
x \sqcap y &= y \sqcap x & x \sqcup y &= y \sqcup x \\
x \sqcap (y \sqcap z) &= (x \sqcap y) \sqcap z & x \sqcup (y \sqcup z) &= (x \sqcup y) \sqcup z \\
x \sqcap (x \sqcup y) &= x \sqcap x & x \sqcup (x \sqcap y) &= x \sqcup x \\
\neg\neg(x \sqcap y) &= x \sqcap y & \lrcorner\lrcorner(x \sqcup y) &= x \sqcup y \\
\neg(x \sqcap x) &= \neg x & \lrcorner(x \sqcup x) &= \lrcorner x \\
x \sqcap \neg x &= \bot & x \sqcup \lrcorner x &= \top \\
\neg\bot &= \top \sqcap \top & \lrcorner\top &= \bot \sqcup \bot \\
\neg\top &= \bot & \lrcorner\bot &= \top \\
x_{\sqcap\sqcup\sqcap} &= x_{\sqcap\sqcup} & x_{\sqcup\sqcap\sqcup} &= x_{\sqcup\sqcap}
\end{aligned}
$$

*where $t_{\sqcap} := t \sqcap t$ and $t_{\sqcup} := t \sqcup t$ is defined for every term t.*

Observe that these properties are exactly those required to define a double Boolean algebra [5]. In intermediate points between preconcepts and concepts, taking constraints more restrictive than the first, but relaxing the latter, are the protoconcepts and semiconcepts.

**Definition 1.** *Let $\mathbb{K} = (G, M, I)$ be a fuzzy formal context. Then,*
1. *A pair $(A, B) \in L^G \times L^M$ is said to be a protoconcept if $A^{\uparrow\downarrow} = B^{\downarrow}$ (or, equivalently, $B^{\downarrow\uparrow} = A^{\uparrow}$).*
2. *A pair $(A, B) \in L^G \times L^M$ is said to be a $\sqcup$-semiconcept if $A = B^{\downarrow}$.*
3. *A pair $(A, B) \in L^G \times L^M$ is said to be a $\sqcap$-semiconcept if $A^{\uparrow} = B$.*

Obviously, every concept is a semiconcept, every semiconcept is a protoconcept, and every protoconcept is a preconcept. These ideas have their significance in FCA. A protoconcept $(A, B)$ is a preconcept that generates only one concept, since $(A^{\uparrow\downarrow}, A^{\uparrow}) = (B^{\downarrow}, B^{\downarrow\uparrow})$. Thus, the protoconcept $(A, B)$ represents the fundamental information about the formal concept it generates. Hence, the idea of a protoconcept is useful for comprehending which conceptual information is transferred from a formal context to some of its contextual extensions.

On the other hand, a semiconcept is a preconcept where one of its components is closed with respect the Galois connection (note that this is not necessarily true for protoconcepts). This implies that, considering a $\sqcup$-semiconcept $(A, B)$, $B$ is a *characteristic attribute set* for the formal concept $(A, B^{\downarrow\uparrow})$, and analogously $A$ is a *characteristic object set* if $(A, B)$ is a $\sqcap$-semiconcept.

From an algebraic point of view, the set of protoconcepts $\mathfrak{P}(\mathbb{K})$, together with the operations $\sqcap$, $\sqcup$, $\neg$, and $\lrcorner$, and the constants $\top$ and $\bot$, is a *double Boolean algebra*, called the *protoconcept algebra*, denoted by $\underline{\mathfrak{P}}(\mathbb{K})$. Furthermore, Vormbrock and Wille [15] proved that each double Boolean algebra is embeddable into some algebra of protoconcepts.

For a deeper comprehension of the algebraic structure of $\underline{\mathfrak{P}}(\mathbb{K})$, the two sets of $\sqcap$- and $\sqcup$-semiconcepts are defined as follows [5,15]:

$$\mathfrak{P}(\mathbb{K})_{\sqcap} := \{(A, A^{\uparrow}) : A \in L^G\} \text{ and } \mathfrak{P}(\mathbb{K})_{\sqcup} := \{(B^{\downarrow}, B) : B \in L^M\}.$$

Then, $\mathfrak{H}(\mathbb{K}) := \mathfrak{P}(\mathbb{K})_{\sqcap} \cup \mathfrak{P}(\mathbb{K})_{\sqcup}$ is the set of all semiconcepts that, when endowed with the (restrictions of the) operations $\sqcap$, $\sqcup$, $\neg$, and $\lrcorner$, and the constants $\top$ and $\bot$, is a subalgebra of $\underline{\mathfrak{P}}(\mathbb{K})$, called the *semiconcept algebra*, and denoted by $\underline{\mathfrak{H}}(\mathbb{K})$, which has the structure of a *pure double Boolean algebra*.

These algebras have attracted much attention, not only because they are generalisations of concept algebra, which is a powerful tool for formal reasoning and knowledge representation, but also because of the possibility of defining logics on them [16,17]. These logics can be used to study properties and relations of concepts in different domains and applications. Moreover, these algebras also allow for exploration and approximation [18,19] by means of *rough sets* [20], which are sets that, intuitively, have vague or imprecise boundaries. Rough sets can help deal with uncertainty and incompleteness in data analysis and decision-making.

## 2.2. L-Fuzzy Algebras and Lattices

Let us recall some basic notions on fuzzy algebras and fuzzy lattices. For this fuzzification, we require that $L$ is a bounded complete lattice, with 0 and 1 denoting its smallest and greatest elements, respectively.

**Definition 2** ([21]). *Let $\mathcal{A} = (A, F)$ be an algebra. A mapping $\overline{A} \colon A \to L$ is said to be an L-fuzzy algebra if, for all $x_1, \ldots, x_n$ and any n-ary operation $f$ in $F$,*

$$\overline{A}(f(x_1, \ldots, x_n)) \geq \overline{A}(x_1) \wedge \cdots \wedge \overline{A}(x_n),$$

*and for every constant $c \in \mathcal{A}$, $\overline{A}(c) = 1$.*

Then, it is easy to check that any $\alpha$-cut of an $L$-fuzzy algebra is an algebra. Recall the idea of an $\alpha$-cut of an arbitrary fuzzy set $R$ over a universe $X$: $R_\alpha := \{x \in X : R(x) \geq \alpha\}$

**Definition 3** ([22]). *Let $(M, \wedge, \vee)$ be a lattice. A mapping $\overline{M} \colon M \to L$ is called a lattice-valued fuzzy lattice (L-fuzzy lattice) if all the p-cuts $(p \in L)$ of $\overline{M}$ are sublattices of M.*

The following characterisation for fuzzy lattices, analogous to the definition given to $L$-fuzzy algebra, will be helpful in the technical results of the next sections.

**Proposition 2** ([22]). *Let $(M, \wedge_M, \vee_M)$ be a lattice. Then a mapping $\overline{M} \colon M \to L$ is an L-fuzzy lattice if, and only if, both of the following relations hold for all $x, y \in M$:*

$$\overline{M}(x \wedge_M y) \geq \overline{M}(x) \wedge \overline{M}(y) \quad and \quad \overline{M}(x \vee_M y) \geq \overline{M}(x) \wedge \overline{M}(y).$$

## 2.3. Graded Subsethood of L-Fuzzy Sets

Since our purpose in this work is to relax the conceptual scheme on preconcepts, protoconcepts, semiconcepts, and concepts, to allow a fuzzy transition between these items and provide a measure of *closeness to conceptuality*, a generalisation of the notion of set inclusion for fuzzy sets is mandatory. In this work, we use as such a generalisation the **graded subsethood** proposed by Bělohlávek [9], for $L$-fuzzy sets over a universe $X$, which is defined as follows:

**Definition 4** ([9]). *Consider a universe X and a complete residuated lattice $(L, \wedge, \vee, \otimes, \to, 0, 1)$. The graded subsethood of $A \in L^X$ into $B \in L^X$ is*

$$S(A, B) := \bigwedge_{x \in X} (A(x) \to B(x)).$$

Observe that, intuitively, $S(A, B)$ expresses the degree to which each element of $A$ is an element of $B$. Note that, $S(A, B)$ is the greatest degree in $L$ such that $S(A, B) \otimes A \subseteq B$.

This degree of inclusion mirrors the main properties of the classical set inclusion. Next, we list those that will be useful in the remainder of this work.

**Theorem 1** ([9]). *Let $A, B, C$ be L-fuzzy sets and $\{D_t\}_{t \in T}$ a family of L-fuzzy sets. Then:*

1. *If $A \subseteq B$, $S(B, C) \leq S(A, C)$.*
2. *If $B \subseteq C$, $S(A, B) \leq S(A, C)$.*
3. *$S(A, \bigcap_t D_t) = \bigwedge_t S(A, D_t)$.*
4. *$S(\bigcup_t D_t, C) = \bigwedge_t S(D_t, C)$.*
5. *$S(A, B) = 1$ if, and only if, $A \subseteq B$.*

In the remainder of this work, we will use the graded $S$ to relax the conditions used in FCA to define its conceptual scheme.

## 3. Fuzzification of Conceptual Structures

In this section, we present the main results regarding the construction of $L$-fuzzy lattices and $L$-fuzzy algebras, extending the classical scheme of conceptual structures. Note that the set $\mathfrak{V}(\mathbb{K})$ of preconcepts, plays an essential role in this discussion: it is both a complete lattice with the usual order $\leq$, and a generalised double Boolean algebra with operators $\sqcap$, $\sqcup$, $\neg$, and $\rightharpoondown$, and constants $\top$ and $\bot$. With these operators, $L^G \times L^M$ is not a double Boolean algebra (it does not fulfill the conditions in Proposition 1). In addition, the sets of protoconcepts and semiconcepts are double Boolean algebras, but not lattices. We can only guarantee this double algebraic structure (being a lattice and a double Boolean algebra) for the preconcept lattice. This fact makes preconcepts an outstanding point in the road to fuzzifying conceptual structures.

### 3.1. Fuzzy Preconcept Lattice

Here, we present the strategy we will follow to assign different degrees of *preconceptuality* to pairs $(A, B) \in L^G \times L^M$. To do this, we will define a mapping $\mathcal{V} : L^G \times L^M \to L$, so that $\mathcal{V}$ is an $L$-fuzzy lattice. Thus, $\mathcal{V}(A, B) \in L$ will represent the degree to which $(A, B)$ can be considered a preconcept. Let us define, for $(A, B) \in L^G \times L^M$,

$$\mathcal{V}(A, B) = S(A, B^\downarrow). \tag{4}$$

Note that its 0-cut coincides with $L^G \times L^M$ and its 1-cut is $\mathfrak{V}(\mathbb{K})$, since

$$\mathcal{V}(A, B) \geq 1 \Leftrightarrow S(A, B^\downarrow) = 1 \Leftrightarrow A \subseteq B^\downarrow \Leftrightarrow (A, B) \text{ is a preconcept.}$$

Note that we could have defined another index $\mathcal{V}' : L^G \times L^M \to L$, by $\mathcal{V}'(A, B) := S(B, A^\uparrow)$. All the discussion below for $\mathcal{V}$ can be carried out analogously for $\mathcal{V}'$, but, as a matter of fact, under the framework of this paper, $\mathcal{V}$ and $\mathcal{V}'$ coincide.

**Lemma 1.** *Given a formal context $\mathbb{K} = (G, M, I)$ and $(A, B) \in L^G \times L^M$. Then*

$$S(A, B^\downarrow) = S(B, A^\uparrow).$$

**Proof.** Let $(A, B) \in L^G \times L^M$, then

$$S(A, B^\downarrow) = \bigwedge_{g \in G} \left( A(g) \to B^\downarrow(g) \right)$$

$$\overset{\text{Equation (1)}}{=} \bigwedge_{g \in G} \left( A(g) \to \left( \bigwedge_{m \in M} B(m) \to I(g, m) \right) \right)$$

$$\overset{(i)}{=} \bigwedge_{g \in G, m \in M} \left( A(g) \to (B(m) \to I(g, m)) \right)$$

$$\overset{(ii)}{=} \bigwedge_{g\in G, m\in M} (B(m) \to (A(g) \to I(g,m)))$$

$$\overset{(iii)}{=} \bigwedge_{m\in M} \left( B(m) \to \left( \bigwedge_{g\in G} (A(g) \to I(g,m)) \right) \right)$$

$$= \bigwedge_{m\in M} \left( B(m) \to A^\uparrow(m) \right) = S(B, A^\uparrow),$$

where $(i)$ holds by (2.51) in [9], $(ii)$ holds by (2.41) in [9], and $(iii)$ holds by (2.51) in [9]. $\square$

**Theorem 2.** *Let* $\mathcal{V} : L^G \times L^M \to L$ *and* $\mathcal{V}' : L^G \times L^M \to L$ *defined by*

$$\mathcal{V}(A, B) = S(A, B^\downarrow) \qquad and \qquad \mathcal{V}'(A, B) = S(B, A^\uparrow).$$

*Then,* $\mathcal{V}(A, B) = \mathcal{V}'(A, B)$, *for all* $(A, B) \in L^G \times L^M$.

**Proof.** It is a direct consequence of Lemma 1. $\square$

To prove that $\mathcal{V}$ is an *L*-fuzzy lattice, we will show that the inequalities in Proposition 2 hold for any collection in $L^G \times L^M$, which is the purpose of the next result.

**Lemma 2.** *Given a collection* $\{(A_t, B_t)\}_{t\in T} \subseteq L^G \times L^M$, *we have:*

1. $\mathcal{V}\left( \bigwedge_t (A_t, B_t) \right) \geq \bigwedge_t \mathcal{V}(A_t, B_t).$

2. $\mathcal{V}\left( \bigvee_t (A_t, B_t) \right) \geq \bigwedge_t \mathcal{V}(A_t, B_t).$

**Proof.** 1. For the first item we have the following,

$$\mathcal{V}\left( \bigwedge_t (A_t, B_t) \right) = \mathcal{V}\left( \bigcap_t A_t, \bigcup_t B_t \right) = S\left( \bigcap_t A_t, \left( \bigcup_t B_t \right)^\downarrow \right)$$

$$\overset{(i)}{=} S\left( \bigcap_t A_t, \bigcap_t B_t^\downarrow \right) \overset{(ii)}{=} \bigwedge_t S\left( \bigcap_t A_t, B_t^\downarrow \right)$$

$$\overset{(iii)}{\geq} \bigwedge_t S\left( A_t, B_t^\downarrow \right) = \bigwedge_t \mathcal{V}(A_t, B_t),$$

where: in $(i)$, we have used $(\bigcup B_i)^\downarrow = \bigcap B_i^\downarrow$ (see [2]); in $(ii)$, the property of Theorem 1 (item 3); and in $(iii)$, the antitonicity of $S$ in the first component, Theorem 1 (item 1).

2. For the second item we have,

$$\mathcal{V}\left( \bigvee_t (A_t, B_t) \right) = \mathcal{V}\left( \bigcup_t A_t, \bigcap_t B_t \right) = S\left( \bigcup_t A_t, \left( \bigcap_t B_t \right)^\downarrow \right)$$

$$\overset{(i)}{=} \bigwedge_t S\left( A_t, \left( \bigcap_t B_t \right)^\downarrow \right)$$

$$\overset{(ii)}{\geq} \bigwedge_t S\left( A_t, B_t^\downarrow \right) = \bigwedge_t \mathcal{V}(A_t, B_t),$$

where: in $(i)$, the property of Theorem 1 (item 4) has been applied; and in $(ii)$, the isotonicity of $S$ in the second component, Theorem 1 (item 2)

Thus, proving the claim. $\quad\square$

We arrive at the main result of this subsection.

**Theorem 3.** *The mapping* $\mathcal{V} : L^G \times L^M \to L$ *is an L-fuzzy lattice.*

**Proof.** This is a direct consequence of the last result and Proposition 2. $\quad\square$

Thus, for every $(A, B) \in L^G \times L^M$, the value $\mathcal{V}(A, B) \in L$ is the degree to which $(A, B)$ can be considered a preconcept.

**Example 1.** *Let us consider the lattice* $L = [0, 1]$, *equipped with the usual linear order and the Gödel structure. Consider the formal context* $\mathbb{K} = (G, M, I)$ *with* $G = \{g_1, g_2, g_3\}$, $M = \{m_1, m_2, m_3\}$ *and* $I \in L^{G \times M}$, *as presented in Table 1.*

**Table 1.** Formal context for the example.

|       | $m_1$ | $m_2$ | $m_3$ |
|-------|-------|-------|-------|
| $g_1$ | 1     | 0.2   | 0     |
| $g_2$ | 0.5   | 1     | 1     |
| $g_3$ | 0     | 0     | 0.7   |

Let us take $(A, B) = (\{^{0.7}/g_2\}, \{m_1\})$. *Observe that* $B^{\downarrow} = \{g_1, ^{0.5}/g_2\}$, *so* $A \nsubseteq B^{\downarrow}$ *and hence* $(A, B) \in L^G \times L^M$ *is not a preconcept. We can compute its preconceptuality index by:*

$$\mathcal{V}(A, B) = S\left(A, B^{\downarrow}\right) = S\left(\{^{0.7}/g_2\}, \{g_1, ^{0.5}/g_2\}\right) =$$
$$= (0 \to 1) \wedge (0.7 \to 0.5) = 0.5.$$

*3.2. Fuzzy Conceptual Algebras*

In this subsection, we will show how to construct an $\mathbb{O}$-fuzzy algebra (for a suitable $\mathbb{O}$) from the double Boolean algebra of preconcepts. Then, each $\alpha$-cut is a subalgebra and, moreover, certain precise cuts coincide with the algebras of protoconcepts, semiconcepts, and concepts. This way, as we mentioned before, the corresponding notions are relaxed, allowing us to speak of degrees of *protoconceptuality*, *semiconceptuality*, and *conceptuality*.

If we recall the successive definitions of preconcepts, protoconcepts, and semiconcepts, we can define a strategy, similar to the one used in the previous subsection for the *L*-fuzzy lattice of preconcepts, to find the desired fuzzy algebra. The idea is to break down the problem into segments: in a first stage, we will study how to construct a fuzzy algebra whose *extremes* are those of preconcepts and protoconcepts; next, we will define a fuzzy algebra between preconcepts and concepts, appropriately so that semiconcepts correspond to certain significant cuts; finally, we will *assemble* these algebras to define a single one, that covers the entire path from preconcepts to concepts.

Before developing the fuzzy algebras, we state the following theoretical result, which will be useful throughout this transition from preconcepts to concepts:

**Lemma 3.** *Let us consider a collection* $\{(A_t, B_t)\}_{t \in T} \subseteq \mathfrak{V}(\mathbb{K})$ *(of at least two preconcepts), then:*

1. $\displaystyle\bigsqcap_t (A_t, B_t)$ *is a* $\sqcap$-*semiconcept.*

2. $\displaystyle\bigsqcup_t (A_t, B_t)$ *is a* $\sqcup$-*semiconcept.*

**Proof.** The proof is straightforward from the definition of $\sqcap$ and $\sqcup$ in Equation (3), and the definition of semiconcepts in Definition 1. $\quad\square$

3.2.1. Fuzzy Algebra as a Measure of *Protoconceptuality*

Let us start with the set $\mathfrak{V}(\mathbb{K})$ of preconcepts, and define a mapping $\mathcal{P} : \mathfrak{V}(\mathbb{K}) \to L$ such that its 1-cut is precisely the set of protoconcepts. For this, we recall the defining property of protoconcepts, that is, a preconcept $(A, B) \in \mathfrak{V}(\mathbb{K})$ must verify $B^{\downarrow} = A^{\downarrow\uparrow}$ (which is equivalent to $A^{\uparrow} = B^{\downarrow\uparrow}$) to be called a protoconcept. Since $(A, B)$ is a preconcept, we already have $B \subseteq A^{\uparrow}$ (and, equivalently, $A \subseteq B^{\downarrow}$). In this case, by the application of the derivation operators, we have that, for preconcepts, $A^{\uparrow\downarrow} \subseteq B^{\downarrow}$ and $B^{\downarrow\uparrow} \subseteq A^{\uparrow}$. Therefore, one only has to check that a preconcept $(A, B)$ verifies $B^{\downarrow} \subseteq A^{\uparrow\downarrow}$ or $A^{\uparrow} \subseteq B^{\downarrow\uparrow}$ to have that $(A, B)$ is also a protoconcept. Hence, given a preconcept $(A, B)$, we define

$$\mathcal{P}(A, B) := S(B^{\downarrow}, A^{\uparrow\downarrow}). \tag{5}$$

Observe that we could define $\mathcal{P}' : \mathfrak{V}(\mathbb{K}) \to L$ by $\mathcal{P}'(A, B) := S(A^{\uparrow}, B^{\downarrow\uparrow})$. As a consequence of Lemma 1, we have that $\mathcal{P}' \equiv \mathcal{P}$.

**Lemma 4.** *Let $(A, B) \in \mathfrak{V}(\mathbb{K})$ be a preconcept and $\{(A_t, B_t)\}_{t \in T} \subseteq \mathfrak{V}(\mathbb{K})$ be a collection of at least two preconcepts. Then:*

1. *$(A, B)$ is a protoconcept if, and only if, $\mathcal{P}(A, B) = 1$.*

2. $\mathcal{P}\left(\bigsqcap_t (A_t, B_t)\right) = \mathcal{P}\left(\bigsqcup_t (A_t, B_t)\right) = 1.$

**Proof.** 1. Let us suppose $(A, B)$ is a protoconcept. Then, it verifies, by Definition 1, $B^{\downarrow} = A^{\downarrow\uparrow}$ hence, particularly, $\mathcal{P}(A, B) = S(B^{\downarrow}, A^{\downarrow\uparrow}) = 1$. Conversely, applying the rationale of the lines above this lemma, we have that if $\mathcal{P}(A, B) = 1$ then $(A, B)$ is a protoconcept.

2. Note that both $\bigsqcap_t (A_t, B_t)$ and $\bigsqcup_t (A_t, B_t)$ are semiconcepts, by application of Lemma 3. Since every semiconcept is a protoconcept, by item 1 above, we have the desired result. □

This last result enables us to prove that $\mathcal{P}$ is an L-fuzzy algebra:

**Theorem 4.** *The mapping $\mathcal{P} : \mathfrak{V}(\mathbb{K}) \to L$ is an L-fuzzy algebra, where the 0-cut is $\mathfrak{V}(\mathbb{K})$ and the 1-cut is the set of protoconcepts, $\mathfrak{P}(\mathbb{K})$. Consequently, $\underline{\mathfrak{P}}(\mathbb{K})$ is an L-fuzzy subalgebra of $\underline{\mathfrak{V}}(\mathbb{K})$.*

**Proof.** We only need to prove that $\mathcal{P}$ is an *L*-fuzzy algebra. By Definition 2, we must prove the two inequalities $\mathcal{P}(\bigsqcap_t(A_t, B_t)) \geq \bigwedge_t \mathcal{P}(A_t, B_t)$ and $\mathcal{P}(\bigsqcup_t(A_t, B_t)) \geq \bigwedge_t \mathcal{P}(A_t, B_t)$ for every collection $\{(A_t, B_t)\}_{t \in T} \subseteq \mathfrak{V}(\mathbb{K})$. This is true, since, by Lemma 4, these two indices are equal to $1 \in L$. Particularly, we have the desired inequalities. □

In this sense, $\mathcal{P}(A, B)$ can be considered as the degree of *protoconceptuality* of $(A, B)$. The closer to 1, the closer $(A, B)$ is to being a protoconcept.

**Example 2.** *Let us continue with our previous Example 1. Let us consider*

$$(A_1, B_1) = \left(\{{}^{0.5}/g_2\}, \{m_2, {}^{0.7}/m_3\}\right).$$

*It is not a protoconcept, since $B_1^{\downarrow} = \{g_2\}$ and $A_1^{\uparrow\downarrow} = \{{}^{0.5}/g_2\}$ are not equal. It is a preconcept, as*

$$\mathcal{V}(A_1, B_1) = S\left(A_1, B_1^{\downarrow}\right) = S\left(\left\{{}^{0.5}/g_2\right\}, \{g_2\}\right) = 0.5 \to 1 = 1,$$

*and its degree of protoconceptuality is*

$$\mathcal{P}(A_1, B_1) = S\left(B_1^{\downarrow}, A_1^{\uparrow\downarrow}\right) = S\left(\{g_2\}, \left\{{}^{0.5}/g_2\right\}\right) = 1 \to 0.5 = 0.5.$$

3.2.2. Fuzzy Algebra as a Measure of *Semiconceptuality*

The motivation for the last step, is to fuzzify the conditions that make a protoconcept be a formal concept. Indeed, a protoconcept $(A, B)$ satisfying $A = B^\downarrow$ and $B = A^\uparrow$, is a formal concept. Thus, we start by defining, given a preconcept $(A, B)$,

$$\mathcal{H}_\sqcap(A, B) := S(B^{\downarrow\uparrow}, B),$$
$$\mathcal{H}_\sqcup(A, B) := S(A^{\uparrow\downarrow}, A), \tag{6}$$

which can be rewritten as $\mathcal{H}_\sqcap(A, B) = S(A^\uparrow, B)$ and $\mathcal{H}_\sqcup(A, B) = S(B^\downarrow, A)$, when $(A, B)$ is a protoconcept. The reason to define $\mathcal{H}_\sqcap$ and $\mathcal{H}_\sqcup$ over the set of preconcepts, will be clear when we build the final fuzzy algebra of preconcepts. Hence, we can characterise the semiconcepts in terms of these new operators:

**Lemma 5.** *Let $(A, B)$ be a protoconcept. Then:*

1.   *$(A, B)$ is a $\sqcap$-semiconcept if, and only if, $\mathcal{H}_\sqcap(A, B) = 1$.*
2.   *$(A, B)$ is a $\sqcup$-semiconcept if, and only if, $\mathcal{H}_\sqcup(A, B) = 1$.*

**Proof.** Let us only prove the first statement, since the second follows an analogous reasoning. Let us suppose that $(A, B)$ is a $\sqcap$-semiconcept. By Definition 1, $B = A^\uparrow$ and, particularly, $\mathcal{H}_\sqcap(A, B) = S(A^\uparrow, B) = 1$. Conversely, suppose $(A, B)$ is a protoconcept, such that $\mathcal{H}_\sqcap(A, B) = S(A^\uparrow, B) = 1$. Then, $A^\uparrow \subseteq B$ and, since $(A, B)$ is also a preconcept, we have that $B \subseteq A^\uparrow$. By antisymmetry, $B = A^\uparrow$, therefore $(A, B)$ is a $\sqcap$-semiconcept.  □

In the next result, we prove the inequalities needed to say that $\mathcal{H}_\sqcap$ and $\mathcal{H}_\sqcup$ are $\Omega$-fuzzy algebras:

**Lemma 6.** *Let $\{(A_t, B_t)\}_{t\in T}$ be a collection of at least two preconcepts. Then:*

1.   $\mathcal{H}_\sqcap\left(\prod_t (A_t, B_t)\right) \geq \bigwedge_t \mathcal{H}_\sqcap(A_t, B_t).$

2.   $\mathcal{H}_\sqcap\left(\bigsqcup_t (A_t, B_t)\right) \geq \bigwedge_t \mathcal{H}_\sqcap(A_t, B_t).$

3.   $\mathcal{H}_\sqcup\left(\prod_t (A_t, B_t)\right) \geq \bigwedge_t \mathcal{H}_\sqcup(A_t, B_t).$

4.   $\mathcal{H}_\sqcup\left(\bigsqcup_t (A_t, B_t)\right) \geq \bigwedge_t \mathcal{H}_\sqcup(A_t, B_t).$

**Proof.**   1.   Since, by Lemma 3, $\prod_t(A_t, B_t)$ is a $\sqcap$-semiconcept, then, applying Lemma 5, we have that $\mathcal{H}_\sqcap(\prod_t(A_t, B_t)) = 1 \geq \bigwedge_t \mathcal{H}_\sqcap(A_t, B_t)$.
2.   Let us consider a collection $\{(A_t, B_t)\}_{t\in T}$ of at least two preconcepts. Then, the following chain of inequalities holds:

$$\mathcal{H}_\sqcap\left(\bigsqcup_t (A_t, B_t)\right) = \mathcal{H}_\sqcap\left(\left(\bigcap_t B_t\right)^\downarrow, \bigcap_t B_t\right) = S\left(\left(\bigcap_t B_t\right)^{\downarrow\uparrow}, \bigcap_t B_t\right)$$

$$\overset{(i)}{=} \bigwedge_t S\left(\left(\bigcap_t B_t\right)^{\downarrow\uparrow}, B_t\right) \overset{(ii)}{\geq} \bigwedge_t S\left(B_t^{\downarrow\uparrow}, B_t\right) = \bigwedge_t \mathcal{H}_\sqcap(A_t, B_t),$$

where: in *(i)*, we have used the property in Theorem 1 (item 3); and in *(ii)*, the antitonicity in the first component of $S$, as per Theorem 1 (item 1).

The proof for items 3 and 4 follow the same rationale as above, therefore we omit it for the sake of readability.  □

**Theorem 5.** *The mappings $\mathcal{H}_\sqcap : \mathfrak{V}(\mathbb{K}) \to L$ and $\mathcal{H}_\sqcup : \mathfrak{V}(\mathbb{K}) \to L$ are L-fuzzy algebras. Furthermore, their 1-cuts correspond to the sets $\mathfrak{P}(\mathbb{K})_\sqcap$ and $\mathfrak{P}(\mathbb{K})_\sqcup$, respectively. Therefore, $\mathfrak{P}(\mathbb{K})_\sqcap$ and $\mathfrak{P}(\mathbb{K})_\sqcup$ are L-fuzzy subalgebras of $\underline{\mathfrak{V}}(\mathbb{K})$. Consequently, $\underline{\mathfrak{H}}(\mathbb{K})$ is an L-fuzzy subalgebra of $\underline{\mathfrak{V}}(\mathbb{K})$.*

**Proof.** This result is a straightforward consequence of Lemmas 5 and 6 and the definition of fuzzy algebra. □

Then, we can view $\mathcal{H}_\sqcap(A, B)$ and $\mathcal{H}_\sqcup(A, B)$ as measures of how close is $(A, B)$ to being considered a $\sqcap$- or $\sqcup$-semiconcept.

**Example 3.** *We continue with the same context as in Example 1. For*

$$(A_1, B_1) = \left( \{^{0.5}/g_2\}, \{m_2, {}^{0.7}/m_3\} \right),$$

*we had proved that it was not a protoconcept, since $\mathcal{P}(A_1, B_1) = 0.5$. We can compute its semiconceptuality indexes as follows:*

$$\mathcal{H}_\sqcap(A_1, B_1) = S\left( B_1^{\downarrow\uparrow}, B_1 \right) = S\left( \{^{0.5}/m_1, m_2, m_3\}, \{m_2, {}^{0.7}/m_3\} \right)$$
$$= (0.5 \to 0) \wedge (1 \to 1) \wedge (1 \to 0.7) = 0,$$

$$\mathcal{H}_\sqcup(A_1, B_1) = S\left( A_1^{\uparrow\downarrow}, A_1 \right) = S\left( \{^{0.5}/g_2\}, \{^{0.5}/g_2\} \right) = 1.$$

*Although $\mathcal{H}_\sqcup(A_1, B_1) = 1$, it is not a $\sqcup$-semiconcept. This is not in contradiction with Lemma 5, since $(A_1, B_1)$ is not a protoconcept.*

*Now, let us take*

$$(A_2, B_2) = \left( \{^{0.7}/g_2\}, \{m_2, {}^{0.7}/m_3\} \right).$$

*Then,*

$$A_2^\uparrow = \{^{0.5}/m_1, m_2, m_3\}, \qquad B_2^\downarrow = \{g_2\},$$
$$A_2^{\uparrow\downarrow} = \{g_2\}, \qquad B_2^{\downarrow\uparrow} = \{^{0.5}/m_1, m_2, m_3\},$$

*hence $(A_2, B_2)$ is a protoconcept. Let us check its protoconceptuality and semiconceptuality indexes $\mathcal{P}(A_2, B_2)$, $\mathcal{H}_\sqcap(A_2, B_2)$ and $\mathcal{H}_\sqcap(A_2, B_2)$:*

$$\mathcal{P}(A_2, B_2) = S\left( B_2^\downarrow, A_2^{\uparrow\downarrow} \right) = S(\{g_2\}, \{g_2\}) = 1 \to 1 = 1,$$

$$\mathcal{H}_\sqcap(A_2, B_2) = S\left( B_2^{\downarrow\uparrow}, B_2 \right) = S\left( \{^{0.5}/m_1, m_2, m_3\}, \{m_2, {}^{0.7}/m_3\} \right)$$
$$= (0.5 \to 0) \wedge (1 \to 1) \wedge (1 \to 0.7) = 0,$$

$$\mathcal{H}_\sqcup(A_2, B_2) = S\left( A_2^{\uparrow\downarrow}, A_2 \right) = S\left( \{g_2\}, \{^{0.7}/g_2\} \right) = 1 \to 0.7 = 0.7.$$

*Let us now consider*

$$(A_3, B_3) = \left( \{g_2, g_3\}, \{^{0.5}/m_3\} \right).$$

*It is easy to see that $A_3 = B_3^\downarrow$, so $(A_3, B_3)$ is a $\sqcup$-semiconcept. We can check its different indexes:*

$$\mathcal{P}(A_3, B_3) = S\left( B_3^\downarrow, A_3^{\uparrow\downarrow} \right) = S(\{g_2, g_3\}, \{g_2, g_3\}) = (1 \to 1) \wedge (1 \to 1) = 1,$$
$$\mathcal{H}_\sqcap(A_3, B_3) = S\left( B_3^{\downarrow\uparrow}, B_3 \right) = S\left( \{^{0.7}/m_3\}, \{^{0.5}/m_3\} \right) = 0.7 \to 0.5 = 0.5,$$
$$\mathcal{H}_\sqcup(A_3, B_3) = S\left( A_3^{\uparrow\downarrow}, A_3 \right) = S(\{g_2, g_3\}, \{g_2, g_3\}) = (1 \to 1) \wedge (1 \to 1) = 1.$$

### 3.2.3. Assembling Fuzzy Algebras

In this part, we will build a fuzzy algebra of preconcepts, where specific and significant cuts correspond to the subalgebras of protoconcepts, semiconcepts, and formal concepts. The construction of such an algebra is based on the concatenation of $\mathcal{P}$, $\mathcal{H}_\sqcap$, and $\mathcal{H}_\sqcup$. Thus, we consider the Cartesian product $\mathbb{O} = L \times L \times L$, equipped with the product order $\leq_\mathbb{O}$, given by:

$$(f_1, g_1, h_1) \leq_\mathbb{O} (f_2, g_2, h_2) \Leftrightarrow f_1 \leq f_2 \text{ and } g_1 \leq g_2 \text{ and } h_1 \leq h_2. \tag{7}$$

Note that, this order provides $\mathbb{O}$ with the structure of a complete lattice, where the infimum and supremum operators are characterised by the component-wise infimum and supremum in $L$. If no confusion arises, we opt to denote $\wedge$ and $\vee$ the corresponding operators in $\mathbb{O}$. The bottom element of $(\mathbb{O}, \wedge, \vee)$ is $\mathbf{0} = (0, 0, 0)$ and its top element is $\mathbf{1} = (1, 1, 1)$.

We have all the ingredients to define a fuzzy algebra of preconcepts that captures the nuances of protoconcepts and semiconcepts and, therefore, makes those notions more flexible.

**Theorem 6.** *The mapping $\mathcal{F} : \mathfrak{V}(\mathbb{K}) \to \mathbb{O}$, such that*

$$\mathcal{F}(A, B) := (\mathcal{P}(A, B), \mathcal{H}_\sqcap(A, B), \mathcal{H}_\sqcup(A, B)), \tag{8}$$

*is an $\mathbb{O}$-fuzzy algebra of preconcepts. Furthermore,*

1. *Its $(1, 0, 0)$-cut is the set of protoconcepts, $\mathfrak{V}(\mathbb{K})$.*
2. *Its $(1, 1, 0)$-cut and its $(1, 0, 1)$-cut correspond to the sets of $\sqcap$- and $\sqcup$-semiconcepts, respectively.*
3. *Its $\mathbf{1}$-cut is the set of formal concepts.*

**Proof.** In order to prove that $\mathcal{F}$ is an $\mathbb{O}$-fuzzy algebra, we need to show that inequalities in Definition 2 hold for arbitrary $\sqcap$ and $\sqcup$ of collections of preconcepts, which is true by the application of Lemmas 4 and 6, and that $\leq_\mathbb{O}$, $\wedge$ and $\vee$ are defined component-wise in $\mathbb{O}$.

Items 1 and 2 are direct consequences of Theorems 4 and 5. Item 3 can be proved by noting that the $\mathbf{1}$-cut corresponds to protoconcepts ($\mathcal{P}(A, B) = 1$) where $\mathcal{H}_\sqcap(A, B) = 1$ and $\mathcal{H}_\sqcup(A, B) = 1$, therefore, by Lemma 5, to $\sqcap$- and $\sqcup$-semiconcepts at the same time, that is, formal concepts. $\square$

Thus, by measuring how close $\mathcal{F}(A, B)$ is to any of these specific values, we can assess to what extent $(A, B)$ could be considered a protoconcept, a semiconcept, or even a formal concept. The conceptual scheme, together with the fuzzy structures built between items, is represented in Figure 1.

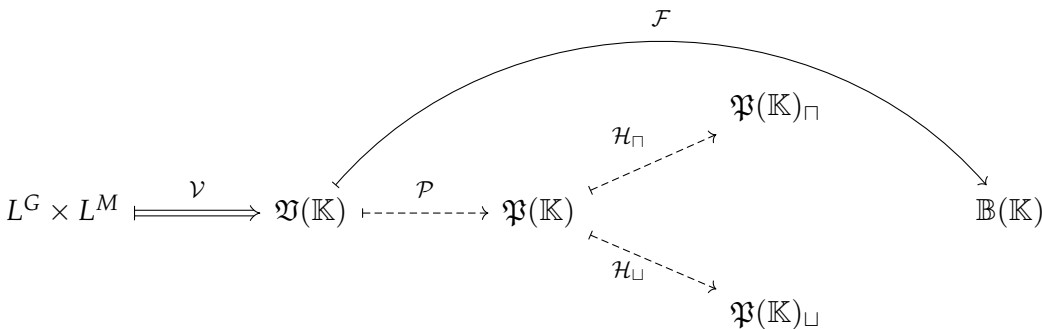

**Figure 1.** *L*-fuzzy lattice and *L*-fuzzy algebras defined in this work. A double solid line stands for the *L*-fuzzy lattice, simple dashed lines represent *L*-fuzzy algebras, and a simple solid line represents the $\mathbb{O}$-fuzzy algebra whose cuts are $\mathfrak{P}(\mathbb{K})$, $\mathfrak{P}(\mathbb{K})_\sqcap$, $\mathfrak{P}(\mathbb{K})_\sqcup$, and the set of formal concepts, $\mathbb{B}(\mathbb{K})$.

In addition, observe that this fuzzy algebra may not be the only one with the specific properties. We can elaborate a result, more general than the previous theorem, that abstracts from the specificity of the above:

**Corollary 1.** *There exist a complete lattice* $(\mathbb{O}, \wedge, \vee, \bot, \top)$; *a mapping* $\mathcal{F} : \mathfrak{V}(\mathbb{K}) \to \mathbb{O}$, *such that* $\mathcal{F}$ *is an* $\mathbb{O}$-*fuzzy algebra of the preconcepts; and constants* $\alpha_1, \alpha_2, \alpha_3, \alpha_4 \in \mathbb{O}$, *such that: the* $\alpha_1$-*cut corresponds to the set of protoconcepts; the* $\alpha_2$- *and* $\alpha_3$-*cuts, to the sets of* $\sqcap$- *and* $\sqcup$-*semiconcepts, respectively; and the* $\alpha_4$-*cut, to the set of formal concepts. Furthermore, we can take* $\alpha_1 = \alpha_2 \wedge \alpha_3$ *and* $\alpha_4 = \alpha_2 \vee \alpha_3 = \top$.

*This fact makes all of* $\underline{\mathfrak{P}}(\mathbb{K}), \underline{\mathfrak{P}}(\mathbb{K})_\sqcap, \underline{\mathfrak{P}}(\mathbb{K})_\sqcup, \underline{\mathfrak{H}}(\mathbb{K}) := \underline{\mathfrak{P}}(\mathbb{K})_\sqcap \cup \underline{\mathfrak{P}}(\mathbb{K})_\sqcup$ *(the* pure *double Boolean algebra of semiconcepts) and* $\underline{\mathbb{B}}(\mathbb{K})$ *fuzzy subalgebras of the preconcept algebra.*

**Example 4.** *Resuming our last examples, consider the same elements*

$$(A_1, B_1) = \left( \{ {}^{0.5}/g_2 \}, \{ m_2, {}^{0.7}/m_3 \} \right),$$
$$(A_2, B_2) = \left( \{ {}^{0.7}/g_2 \}, \{ m_2, {}^{0.7}/m_3 \} \right),$$
$$(A_3, B_3) = \left( \{ g_2, g_3 \}, \{ {}^{0.5}/m_3 \} \right).$$

*We have already computed* $\mathcal{P}(A_1, B_1) = 0.5$, $\mathcal{H}_\sqcap(A_1, B_1) = 0$ *and* $\mathcal{H}_\sqcup(A_1, B_1) = 1$. *It is a proper preconcept, since it is not a protoconcept. Its conceptuality index is* $\mathcal{F}(A_1, B_1) = (0.5, 0, 1)$.

*We also computed* $\mathcal{P}(A_2, B_2) = 1$, $\mathcal{H}_\sqcap(A_2, B_2) = 0$ *and* $\mathcal{H}_\sqcup(A_2, B_2) = 0.7$. *Thus, in the full* $\mathbb{O}$-*fuzzy algebra,* $(A_2, B_2)$ *has a conceptuality index of* $\mathcal{F}(A_2, B_2) = (1, 0, 0.7)$. *Therefore, we can say it is closer to being a* $\sqcup$-*semiconcept than to a* $\sqcap$-*semiconcept, but it is neither, since* $(A_2, B_2)$ *is a proper protoconcept.*

*For* $(A_3, B_3)$, *we have* $\mathcal{P}(A_3, B_3) = 1$, $\mathcal{H}_\sqcap(A_3, B_3) = 0.5$ *and* $\mathcal{H}_\sqcup(A_3, B_3) = 1$. *This means that* $\mathcal{F}(A_3, B_3) = (1, 0.5, 1)$. *As is natural,* $(A_3, B_3)$ *is closer to being a concept than* $(A_2, B_2)$, *as we can check by noting that* $\mathcal{F}(A_2, B_2) <_{\mathbb{O}} \mathcal{F}(A_3, B_3)$.

## 4. Conclusions

This paper has investigated the different rich algebraic conceptual structures in formal concept analysis, where we know that the fuzzy powerset lattice, the most general set, contains several interesting formations, such as formal concepts and their generalisations. Our main contributions are:

1. A gradation of the powerset lattice, that preserves the complete sub-lattice property for all $\alpha$-cuts and recovers the complete lattice of preconcepts for the 1-cut.

2. An $L$-fuzzy algebra of preconcepts, that preserves the subalgebra structure for all $\alpha$-cuts and recovers the algebra of protoconcepts for the 1-cut. Moreover, this gradation can be extended to include both types ($\sqcap$- and $\sqcup$-) of semiconcepts as particular cases for different $\alpha$-cuts.

3. Using the complete lattice $L \times L \times L$, induce an $L \times L \times L$-fuzzy algebra of preconcepts, which is able to recover the algebras of protoconcepts, $\sqcup$-semiconcepts, $\sqcap$-semiconcepts, and formal concepts for different combinations of $\alpha$-cuts: the $(1, 0, 0)$-cut is the algebra of protoconcepts, the $(0, 1, 0)$-cut is the set of $\sqcap$-semiconcepts, the $(0, 0, 1)$-cut is the set of $\sqcup$-semiconcepts, and the $(1, 1, 1)$-cut is the set of formal concepts.

With this gradation in conceptual structure, we are able to determine the degree of (pre-, proto-, and semi-) conceptuality, thus allowing us to know how *close* a pair $(A, B)$ is to being one of the crisp conceptual structures defined in the classical setting, hence this framework is able to capture and model situations where ambiguity and imprecision in data are present.

These contributions extend previous results on the gradation of the powerset lattice [7], providing a more comprehensive study of the intermediate algebraic structures that arise between the preconcepts and the formal concepts. In this sense, our work may help in the

understanding of the underlying relationship between the crisp notions of pre-, proto-, and semiconcepts, as well as in providing a panoramic view of the conceptual space in the fuzzy paradigm.

Our findings provide new insights into the formal concept analysis framework and pave the way for future research. One area for further exploration, is the extension of this framework to consider the *extensionality* and *intensionality* of a concept (or any of the intermediate structures) differently. We plan to investigate how, and under what circumstances, other fuzzy algebraic structures might capture the different behaviour of objects and attributes, thus extending the framework beyond the current limits. In addition, we aim to address the issue of fuzzy inclusion measures, and study how they can be used to capture the degree of similarity between sets in a more general setting.

It would also be useful to explore how the gradation can be used for knowledge discovery and representation tasks, such as clustering, classification or ontology learning. Moreover, it would be worthwhile to study how the gradation can be combined with other extensions or variations of FCA, such as rough sets or triadic concepts.

In conclusion, our study contributes to the development of formal concept analysis, by providing a more detailed understanding of its algebraic structures and their relationships. The proposed framework opens up new avenues for future research and provides a valuable tool for analysing complex data structures.

**Author Contributions:** Conceptualization, M.O.-H., D.L.-R. and P.C.; methodology, M.O.-H., D.L.-R. and P.C.; formal analysis, M.O.-H., D.L.-R. and P.C.; investigation, M.O.-H., D.L.-R. and P.C.; resources, M.O.-H., D.L.-R. and P.C.; writing—original draft preparation, M.O.-H., D.L.-R. and P.C.; writing—review and editing, M.O.-H., D.L.-R. and P.C.; visualization, M.O.-H., D.L.-R. and P.C.; supervision, P.C.; project administration, P.C.; funding acquisition, M.O.-H. and P.C. All authors have read and agreed to the published version of the manuscript.

**Funding:** This research was partially supported by the Spanish Ministry of Science, Innovation, and Universities (MCIU), grant number FPU19/01467, and by the State Agency of Resaearch (AEI), the European Social Fund (FEDER), and the Spanish Ministry of Science, Innovation, and Universities (MCIU), grant number PID2021-127870OB-I00.

**Data Availability Statement:** Our manuscript has no associated data.

**Conflicts of Interest:** The authors declare no conflict of interest.

## Abbreviations

The following abbreviations are used in this manuscript:

| | |
|---|---|
| MDPI | Multidisciplinary digital publishing institute |
| DOAJ | Directory of open access journals |
| FCA | Formal concept analysis |

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
