# Peer review of "Fuzzy Algebras of Concepts"

_axioms, doi:10.3390/axioms12040324_

Round 1

Reviewer 1 Report

Formal Concept Analysis is a powerful method of data analysis which provides the construction of special bi-clusters, called formal concepts. This method is applied in various applications.

The authors wrote several fruitful and interesting papers on Formal Concept Analysis and its properties.

In this paper, as the continuation of their results, the authors proposed a fuzzy framework in which the notions of pre-concept, proto-concept, semi-concept and formal concept are unified. These notions correspond to different degrees of generalization. Moreover, they incorporated the fuzziness in their results. They mentioned the results provided by Šostaks et al. (2021) in the Axioms journal, which are connected with their paper, as well.

I appreciate Figure 1 which represents the L-fuzzy lattice and L-fuzzy algebras defined in their work. Moreover, I appreciate four illustrative examples of their novel results. They are based on a formal context introduced in Table 1. The authors proved 6 theorems and several lemmas which correspond to a gradation of the powerset lattice.

The small remarks and comments:

Page 6: it does not fullfil  ….. it does not fulfill

Congratulation to a well-performed research which is a very worthy for a publication.

I recommend accepting the paper in present form.

Author Response

Please, see the attached document.

Reviewer 2 Report

This paper is written in good form and the results seem to be sound and relevant. It can be
accepted after some minor correction.
Some suggestions are given below:
1. Write abstract in a more claiming manner by highlighting your contribution.
2. Please recheck all your derivations meticulously.
3. Have you used any software for evaluation of the derived or proposed equations?
4. Figures etc. should be made more clear and visible.
5. Please correct all grammatical errors.
6. Please check the reference list and cite some papers related to this work.
7. Rewrite conclusion in a precise way.

Author Response

Please, see the attached document.

Reviewer 3 Report

The paper presents a study of the different types of preconcept in Formal Concept Analysis and their relationship with formal concepts. The authors provide a gradation of the fuzzy powerset lattice and the algebra of preconcepts such that different cuts give the distinct protoconcept, semiconcept, and concept algebras. The main contribution of the paper is the degree of (pre, proto and semi)-conceptuality, which allows determining how close a pair (A, B) is to being any of the crisp conceptual structures as defined in the classical setting. Following are the suggestions to improve the quality of paper. 1. One limitation of the paper is that it is highly technical, which may make it difficult for non-experts to understand. Additionally, while the authors provide a clear explanation of their results and should add some example 2. They do not provide much context for why their findings are significant or how they contribute to the broader field of Formal Concept Analysis. 3. The literature review misses many new contribution in this topic for example https://doi.org/10.3390/sym14102084 4. The abstract is poorly written, must be improved 5. The authors should add their major contribution pointwise at the of introduction section. 6. A comprehensive discussion is recommended before conclusion. 7. Finally, the authors should carefully proofread the manuscript to ensure that it is free of errors in spelling, grammar, and punctuation

Author Response

Please, see the attached document.

Round 2

Reviewer 3 Report

Well done